# Magnetic droplet soliton pairs

S. Jiang [1,2,9], S. Chung [2,3,9] ✉, M. Ahlberg [2,9] ✉, A. Frisk [2], R. Khymyn [2], Q. Tuan Le[2,4], H. Mazraati[4], A. Houshang[2], O. Heinonen[5,8] & J. Åkerman [2,4,6,7] ✉

We demonstrate magnetic droplet soliton pairs in all-perpendicular spin-torque nano-oscillators (STNOs), where one droplet resides in the STNO free layer (FL) and the other in the reference layer (RL). Typically, theoretical, numerical, and experimental droplet studies have focused on the FL, with any additional dynamics in the RL entirely ignored. Here we show that there is not only significant magnetodynamics in the RL, but the RL itself can host a droplet driven by, and coexisting with, the FL droplet. Both single droplets and pairs are observed experimentally as stepwise changes and sharp peaks in the dc and differential resistance, respectively. While the single FL droplet is highly stable, the coexistence state exhibits high-power broadband microwave noise. Furthermore, micromagnetic simulations reveal that the pair dynamics display periodic, quasi-periodic, and chaotic signatures controlled by applied field and current. The strongly interacting and closely spaced droplet pair offers a unique platform for fundamental studies of highly non-linear soliton pair dynamics.

Solitons are particle-like solutions to non-linear wave equations and emerge in various physical systems, e.g., shallow water[1], optical fibers[2], conducting polymers[3], Bose-Einstein condensates[4], and magnetic materials[5,6]. In magnetic materials, different spin wave dynamics can be excited by spin-transfer-torque (STT), spin-orbit-torque (SOT), and/or voltage-controlled magnetic anisotropy (VCMA) in spintronic nanodevices[7–13]. For example, using nano-oscillators, rich dynamics can be generated, not only propagating spin waves[14], but also localized solitons, such as bullets[15,16], magnetic droplets[7], vortices[17], and skyrmions[18]. Their highly tunable dynamics are essential for applications in radio-frequency electronics[19,20], magnetic random access memory (MRAM)[21], magnonics[22], neuromorphic computing[23,24], and Ising machines[25].

Practical applications aside, the mathematical framework of soliton physics go by many names, e.g., the nonlinear Schrödinger, the sine-Gordon, or the Landau–Lifshitz–Gilbert equations. Nevertheless, they are all closely related[26]. A magnetic soliton thus represents a specific solution to a very general mathematical problem, and a closer understanding of one specific soliton also advances the insight into others in disparate physical systems.

The magnetic droplet soliton has been the subject of numerous studies, covering theoretical[27–30], numerical[30–34], and experimental aspects[7,35–39]. While droplets can be created in devices with a single ferromagnetic (FM) layer by utilizing the spin Hall effect[40] and, in simulations, using voltage-controlled magnetic anisotropy[41], they have mostly been examined using spin-torque nano-oscillators (STNOs), which contain two magnetic layers. In STNOs an electrical current run through a stack comprising a soft FM layer, a non-magnetic spacer, and a hard FM layer. The electrons are polarized by the hard reference layer, and the resulting spin current counteracts the damping in the soft free layer in which a droplet forms. Earlier studies have generally focused only on the easily excited magnetodynamics of the free layer, neglecting any dynamics in the reference layer. On the other hand, problems with back-hopping in magnetic switching have been

[1]School of Microelectronics, South China University of Technology, 511442 Guangzhou, China. [2]Physics Department, University of Gothenburg, 412 96 Gothenburg, Sweden. [3]Department of Physics Education, Korea National University of Education, Cheongju 28173, Korea. [4]Department of Applied Physics, School of Engineering Sciences, KTH Royal Institute of Technology, 100 44 Stockholm, Sweden. [5]Materials Science Division, Argonne National Laboratory, Lemont, IL 60439, USA. [6]Center for Science and Innovation in Spintronics, Tohoku University, 2-1-1 Katahira, Aoba-ku, Sendai 980-8577, Japan. [7]Research Institute of Electrical Communication, Tohoku University, 2-1-1 Katahira, Aoba-ku, Sendai 980-8577, Japan. [8]Present address: Seagate Technology, 7801 Computer Ave., Bloomington, MN 55435, USA. [9]These authors contributed equally: S. Jiang, S. Chung, M. Ahlberg. ✉e-mail: sjchung76@knue.ac.kr; martina.ahlberg@physics.gu.se; johan.akerman@physics.gu.se

addressed for more than a decade[42]. Back-hopping occurs when the applied dc current is substantially higher than the critical switching current, which results in the destabilization of the reference layer (RL) and excitation of dynamics[43–47]. RL modes have also been observed in magnetic tunnel junctions[48]. Therefore, both the free and the reference layers are expected to demonstrate dynamics when the applied current is sufficiently high.

In this work we use nanocontact STNOs with strong perpendicular magnetic anisotropy (PMA) in both magnetic layers. The layer materials and stack order are shown in Fig. 1a. The device layout allows for the creation of a stable ordinary droplet in the free layer, illustrated in Fig. 1b. Furthermore, the all-perpendicular symmetry also opens up for droplet nucleation in the RL. We apply large current densities to excite significant dynamics in the reference layer and investigate the limit of the RL acting as simply a static polarization layer. We observe clear transitions at high currents and show that both the RL and FL can sustain droplets, which coexist as depicted in Fig. 1c. The droplet pair constitutes a previously unexplored segment in the STNO current-field phase diagram, which we examine by experiments and simulations.

## Results and discussion

### Experimental characteristics of droplet soliton pairs

Droplet nucleation is often identified by a sharp drop in the detected microwave frequency[7,36]. However, in our measurements, the applied field is directed along the PMA axis of both layers. As a consequence, the STNO resistance is unaffected by the in-plane magnetization direction, and it is therefore not possible to use GMR to experimentally harvest the microwave frequency precession of the droplet perimeter. Instead, we use the absolute ($R_{dc}$) and differential resistance ($R_{dV/dI}$) together with the power spectral density (PSD) to identify the different phases. A similar approach was employed in an earlier study where we observed the transition from a droplet to a static bubble[39].

Fig. 2 demonstrates the experimental features of all magnetic states observed: simple static parallel (P) and anti-parallel (AP) alignments of the FL and RL, single FL droplet (D), single FL droplet in the AP state (AP-D), and the novel states of droplet soliton pairs in the P (DP) and the AP (AP-DP) states. A positive applied current ($I$) has no effect on parallel layers (Fig. 2a), since the STT adds to the damping-like torque in this case, giving a virtually constant resistance. When the current polarity is reversed, an ordinary droplet is nucleated at $I \approx -4$ mA (current density $j = -1.4 \times 10^8$ A/cm$^2$), which is manifested by a clear step in $R_{dc}$ and a weak microwave signal[36,37] in a narrow current region just after nucleation. The resistance remains high for a range of about 10 mA without any concomitant noise, which means that the droplet is remarkably stable.

A droplet soliton pair, i.e., the emergence of a coexisting droplet in the reference layer, appears as the current is further increased. The transition is identified by a dramatic onset of low-frequency noise, while the resistance decreases. The microwave noise is likely caused by droplet annihilation/re-nucleation and drift, indicating much less

stability of the DP state. In addition, if both droplets were stable, the resistance should approach the parallel state value. Instead, it levels out at about 17 mΩ, approximately halfway between the P and D states.

We now turn to the opposite polarity of the field, $\mu_0 H = -0.2$ T (Fig. 2b). At this field, the sample is in an AP state, and droplet nucleation is facilitated by positive currents. Although the sweep starts at $I = 21$ mA ($j = 7.4 \times 10^8$ A/cm$^2$), we first describe the results at lower currents. The resistance shows a small drop at $I = 6.8$ mA ($j = 2.4 \times 10^8$ A/cm$^2$), corresponding to droplet formation. The transition is also accompanied by measurable microwave noise, although it is hard to discern. This small signal is highlighted in Fig. S2 in the Supplementary Materials, which present the PSD on a logarithmic scale. The decrease in resistance ($\Delta R_D^{AP} \approx 6$ mΩ) is lower than the corresponding increment of $R$ for a droplet in a parallel setting ($\Delta R_D^P$ *approx* 40 mΩ). The difference in $\Delta R_D$ translates into a difference in size, and the smaller size of the AP-droplet is consistent with earlier results[37]. The device geometry gives rise to a lateral current component[49] that generates an in-plane Zhang-Li torque (ZLT)[50,51], which exerts inward (outward) pressure on the droplet perimeter for positive (negative) currents. Consequently, AP-droplets are smaller than their parallel state counterparts. The AP-droplet becomes unstable at higher currents of about 10 mA ($j = 3.5 \times 10^8$ A/cm$^2$), and ultimately a second droplet nucleates in the reference layer (AP-DP) at 13 mA ($j = 4.6 \times 10^8$ A/cm$^2$) as indicated by the falling resistance and the high noise level.

The anti-parallel state is stable for low negative currents, but at a threshold current the magnetic state switches from AP to P. Since the RL magnetization is anti-parallel to both the FL[45] and the applied field, this layer becomes increasingly unstable with increasing current and finally reverses. The reversal is observed by a marked decline in $R_{dc}$, and a stable ordinary droplet forms without any concomitant microwave signal. The behavior at higher negative currents is identical to $\mu_0 H = +0.2$ T (Fig. 2a). The single droplet phase is followed by RL droplet nucleation (DP), and the backward sweep (orange line) repeats the same features. However, at the highest positive current (very last data point, $I = 21$ mA) the STT forces an anti-parallel alignment, and an AP-droplet pair once again appears.

The results reveal large differences in the experimental traces of droplets in the parallel state compared to the AP state. In a first approximation, they should be the same. However, the in-plane component of the current causes a ZLT, which not only causes unequal droplet sizes but also breaks the symmetry of the system. This influences the evolution in the droplet pair regime. The effect on the observed PSD is remarkable. The maximum power of the P-DP is roughly twice as high compared to the AP-DP, as seen by comparing the dynamic signal in Fig 2b at negative and positive currents. The shape of the signal is also different. Examples of the signal at individual currents are found in Fig. S3 in the Supplementary Materials. At negative currents, the power peaks at low $f$ (0–1 GHz) and decreases with frequency above the peak. The maximum moves towards higher $f$

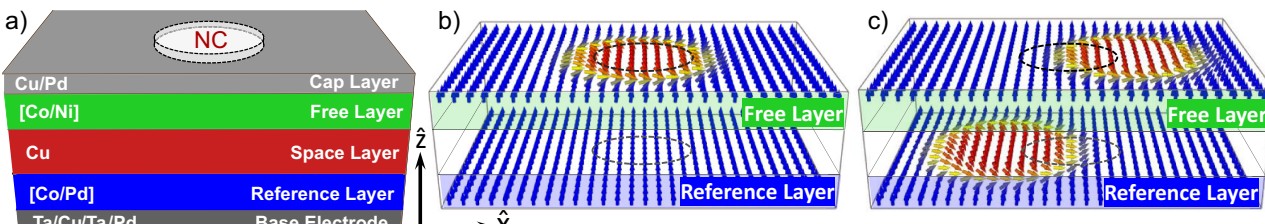

**Fig. 1 | Sample schematics. a** Schematic of an STNO device with stack information. Both the free (Co/Ni multilayer) and the reference layer (Co/Pd multilayer) have strong perpendicular magnetic anisotropy along the $\hat{z}$-axis. The applied current runs through the nanocontact (NC) down to the base electrode, where it flows laterally before it continues back up to two top electrodes located about 5 μm to the left and right of the NC (see Fig. S1 in the Supplementary Materials). **b** Single droplet nucleated in the free layer. **c** Droplet soliton pair residing in both the free and reference layer. Dashed circles indicate the NC areas.

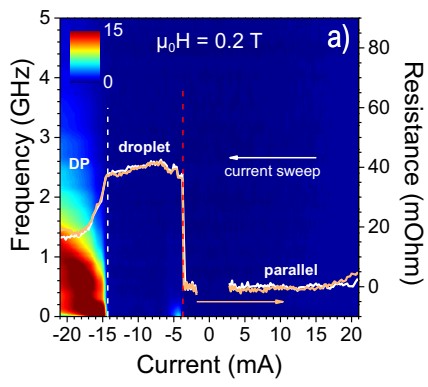
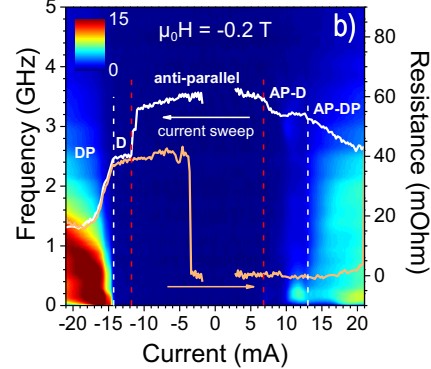

**Fig. 2 | DC and RF characterization.** The dc resistance (white line) and corresponding PSD (color map) of a current sweep from $I = 21$ mA to $I = -21$ mA ($j = 7.4$ to $-7.4 \times 10^8$ A/cm$^2$), at **a** $\mu_0 H = +0.2$ T and **b** $\mu_0 H = -0.2$ T. The orange lines show the resistance of the backward sweeps $I = -21 \to 21$ mA; the corresponding PSDs are not shown. AP, D, and DP denote Anti-Parallel, single Droplet, and Droplet Pair,

respectively. A constant device resistance and a parabolic background (caused by Joule heating) have been removed from the raw data, thus the resulting parallel resistance is zero. Raw resistance data is presented in Fig. S4 in the Supplementary Materials.

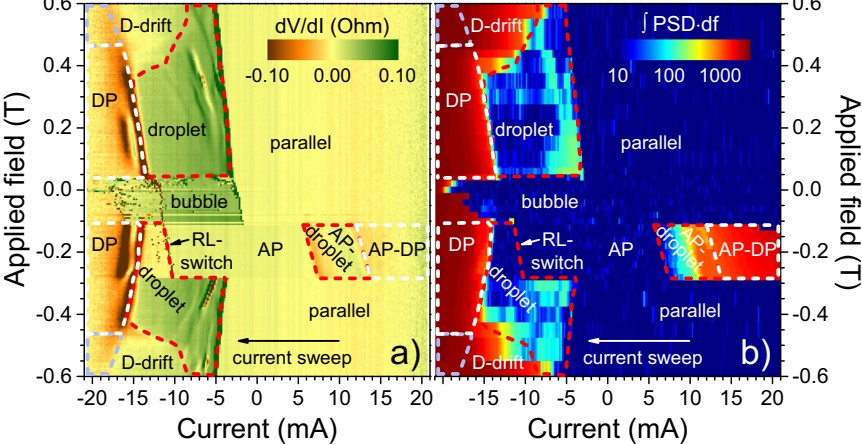

**Fig. 3 | Phase diagrams based on differential resistance and microwave noise.** **a** STNO differential resistance (dV/dI) as a function of applied current and field; a parabolic background has been removed. **b** Microwave noise, integrated over 0–5 GHz, presented on a logarithmic scale. The red lines mark regions of stable single droplets, while areas with droplet pairs (DP) are indicated by white lines. The light blue lines denote sectors where it is hard to distinguish single droplet drift (D-drift) and droplet pair dynamics. Droplets in the anti-parallel (AP) state are found at

positive currents. The plain AP state transforms into a droplet state at negative currents when the reference layer (RL) magnetization switches direction. Between each current sweep from 21 to −21 mA (displayed) and back again (not shown), the sample was first saturated in a field of 0.6 T, whereafter the measurement field was set. The data in (**a** and **b**) are taken from two different measurements. Figure S5 in the Supplementary Materials presents the dV/dI in a without background correction, together with raw differential resistance at five representative fields.

with increasing current. At positive currents, the signal shows a peak around 0.1 GHz for all currents above ≈10 mA ($j = 3.5 \times 10^8$ A/cm$^2$). At higher currents, $I < 15$ mA ($j = 5.3 \times 10^8$ A/cm$^2$), the power is comparable to the initial peak for a range of frequencies, while above roughly 2 GHz the signal progressively weakens.

In short, the signal shape of the droplet pair is different for the P and AP states, and the signal is determined by the DP motion. Consequently, the drift motion of coexisting droplets depends on the initial orientation of the magnetic layers. Nevertheless, the signal of both conditions is similar for frequencies above ≈3 GHz, which implies comparable dynamics on the shortest time scales (largest $f$).

### Current-field phase diagram

The phase diagram displayed in Fig. 3a is constructed from $R_{\mathrm{dV/dI}}$, and Fig. 3b presents microwave noise, integrated over 0–5 GHz, using a logarithmic scale to highlight small signals. The integration is made by summing up the measured power in dB (over the noise floor). Consequently, the relation to true power is lost, and the unit is undefined. Nevertheless, this approach makes it possible to display

small and large signals within the same plot, whereas minor peaks are indistinguishable if the linear scale power [W/Hz] is integrated. Different magnetic states are easily identified by comparing the two subfigures.

The only visible features in positive currents are found in the anti-parallel configuration ($-0.28$ T $< \mu_0 H < -0.10$ T), where a droplet pair nucleates immediately as a large $+I$ is applied. As the current is reduced, the RL droplet first vanishes, followed by the annihilation of the single FL droplet, leaving a simple AP state. In this state, negative currents destabilize the RL[45,52], and at around $I = -10$ mA ($j = -3.5 \times 10^8$ A/cm$^2$) the RL switches into the direction of the applied field. The differential resistance does not reveal the subsequential nucleation of a droplet, and there is no associated microwave noise. However, the magnitude of $R_{\mathrm{dc}}$ after the switch is identical to an ordinary droplet (Fig. 2a) and we conclude that the free layer indeed hosts a droplet once the RL is reversed. Further increase of the current nucleates a droplet in the RL as well, and this transition is clearly visible in Fig. 3a as negative peaks, and in Fig. 3b as a distinct onset of strong noise (marked by white lines).

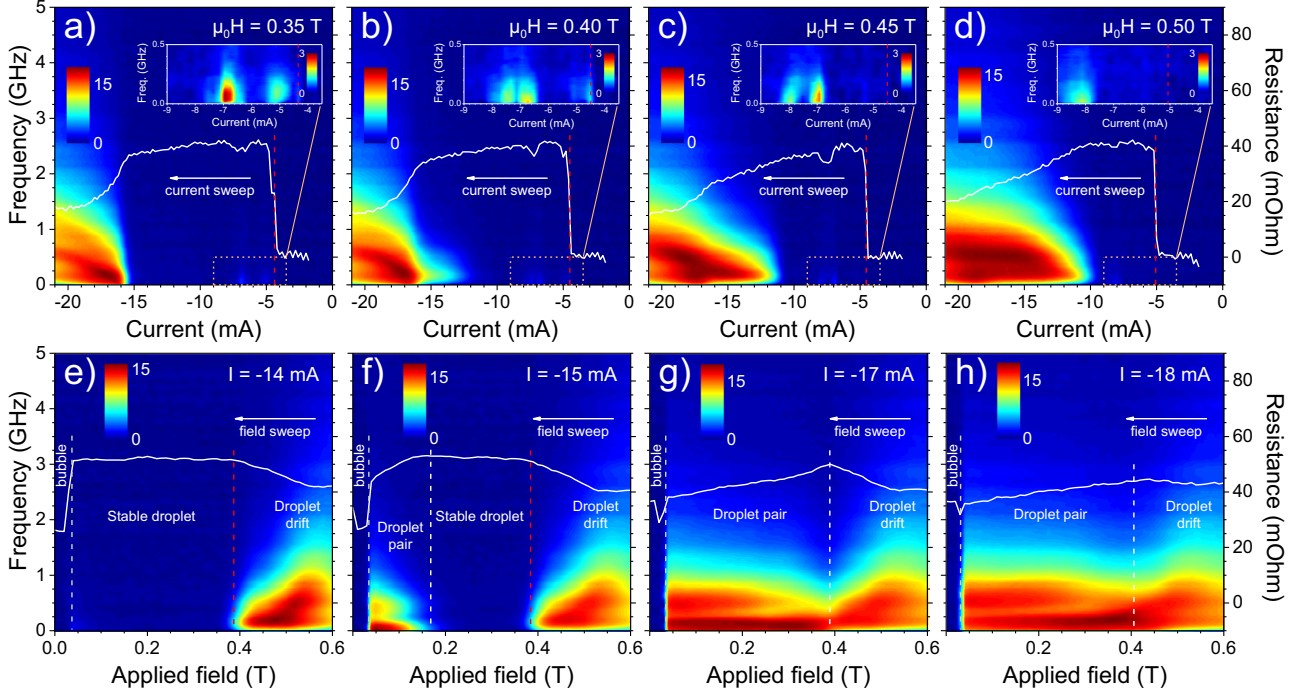

**Fig. 4 | Microwave noise PSD and dc resistance at selected fields and currents.** **a–d** PSD (color map) and dc resistance (white lines) of current sweeps at four different fields: **a** $\mu_0 H = 0.35$ T, **b** $\mu_0 H = 0.40$ T, **c** $\mu_0 H = 0.45$ T, and **d** $\mu_0 H = 0.50$ T. The insets zoom in on small signals in the PSD; the displayed area is given by the orange dashed box. (Large versions of the insets are found in Fig. S6 in the Supplementary Materials). The red dashed lines mark the nucleation of a single droplet.

The current was swept from $I = +21$ mA to $I = -21$ mA, but only data for negative currents is shown. **e–h** PSD (color map) and dc resistance (white lines) of field scans at four different currents: **e** $I = -14$ mA ($j = -5.0 \times 10^8$ A/cm²), **f** $I = -15$ mA ($j = -5.3 \times 10^8$ A/cm²), **g** $I = -17$ mA ($j = -6.0 \times 10^8$ A/cm²), **h** $I = -18$ mA ($j = -6.4 \times 10^8$ A/cm²). A parabolic background is removed from the resistance.

The single droplet nucleation boundaries follow the linear dependence expected by STT-theory[53]. Figure 3 shows that the droplet is exceedingly stable in a wide range of fields and currents, which we have indicated by red lines. The small microwave noise signal observed in this range is related to droplet mode conversions[54] also visible as peaks in $R_{dV/dI}$. The range where a droplet pair is unambiguously present is marked by white dashed lines in Fig. 3. The transition is more blurred above $\mu_0|H| \approx 0.46$ T, and the FL droplet is also accompanied by strong microwave noise at high fields and currents. The range where it is difficult to distinguish single droplet and droplet pairs is indicated by light blue lines.

## Microwave noise characteristics

Fig. 4 shows the dc resistance and microwave noise at four selected fields and currents. The behavior at $\mu_0 H = 0.35$ T (Fig. 4a) is similar to the observation at $\mu_0 H = 0.20$ T (Fig. 2a). The distinct step in resistance is a sign of the nucleation of a single droplet, which remains stable for a wide current range until a droplet pair appears at $I \approx -15$ mA as revealed by $R_{dc}$ and the strong noise. In Fig. 4b the field is increased by 0.05 T and the image changes slightly. The microwave noise appears at a weaker current and covers initially lower frequencies compared to the droplet pair signal. The overall resistance profile remains virtually unchanged. It is still easy to find a distinction between the single and DP phases, and we attribute the initial noise to single droplet drift. As the field is further increased (Fig. 4c, d), the resistance acquires a noticeable slope, and noise appears at even weaker $I$. The distinction between single and paired droplets is smeared out.

The field sweeps in Fig. 4e–h give a more explicit illustration of the single and paired droplets' characteristics and the gradual fading of the border between them. At moderate currents ($I = -14$ mA, Fig. 4e), there are only FL droplets, which freeze into static bubbles close to zero field[39]. The droplet experiences drift at high fields, manifested by

a reduction of the time-averaged resistance and the presence of microwave noise. A comparable drift is observed at the same fields for $I = -15$ mA, but Fig. 4f also unveil the difference between drift noise compared to droplet pair noise. The dynamic signal of coexisting droplets covers a lower frequency range and has a two-peak-like shape, while single droplet drift noise diminishes close to zero GHz. Furthermore, $R_{dc}$ has a positive slope in the DP regime, stays constant for a stable ordinary droplet, and descends in presence of drift. Figure 4g shows that the different features are still visible at $I = -17$ mA, although there is no clear cut between one and another. For even higher currents (Fig. 4h), no well-defined aspect defines the two regimes, although the close to-zero frequency intensity decline at higher fields, indicating single droplet drift.

The insets in Fig. 4a–d zoom in onto selected currents and emphasize low-noise signals in the single droplet phase. The nucleation of a droplet is not always accompanied by measurable dynamics. However, transitions between (single) droplet modes are discernable by noticeable kinks in $R_{dc}$ together with evident low-frequency signals. These mode transitions are also found in Fig. 3 and the associated noise has no intensity above $\approx 0.5$ GHz. The lateral dynamics of mode transitions is consequently much slower compared to the movements related to single droplet drift or droplet pair interactions.

## Micromagnetic simulations

We have performed micromagnetic simulations to further explore the droplet soliton pair phase. Magnetodynamics in both the free and reference layers were simulated simultaneously. A drawback of the used MuMax3[55] package is that the polarization (fixed) layer magnetization is constant by default, which means that back-hopping effects are ignored. To include these effects, we considered the real-time magnetization of one layer to serve as the polarization layer of the other layer. Zero temperature, $T = 0$ K, was used in all simulations.

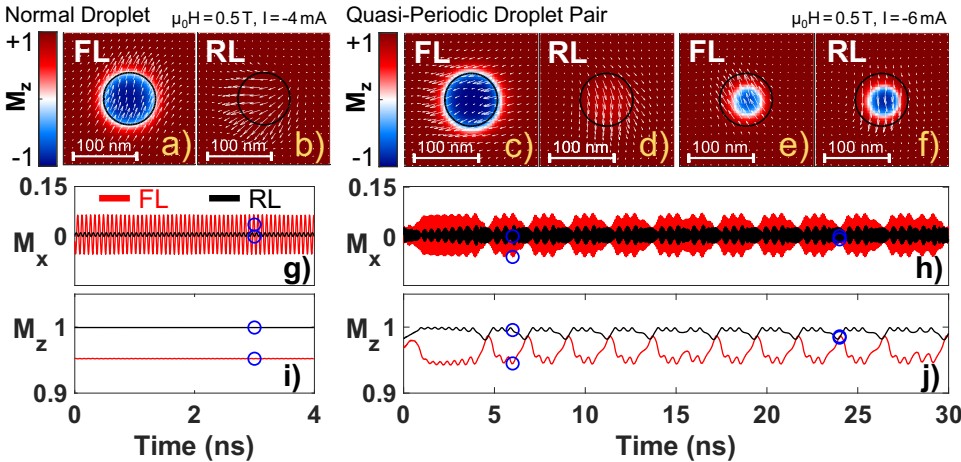

**Fig. 5 | Ordinary free layer droplet and quasi-periodic droplet pair. a–f** Droplet snapshots together with **g–j** the time evolution of $M_x$ (in-plane) and $M_z$ (out-of-plane). The magnetization of the free layer (FL) and the reference layer (RL) are given by red and black lines, respectively. The blue circles in **g–j** show the corresponding time and magnetization of the snapshots.

Overall, the outcome confirms the experimental observation of droplets in the RL. Furthermore, the droplet pair state can, based on its dynamic signature, be divided into three subcategories—periodic, quasi-periodic, and chaotic.

An ordinary droplet is formed at high field and low current ($\mu_0 H = 0.5$ T, $I = -4$ mA ($j = -0.8 \times 10^8$ A/cm$^2$)) as illustrated in Fig. 5a, b, which show snapshots of the magnetic states in the free and reference layers, respectively. In addition, the $M_x$- and $M_z$-components of the two layers are displayed as a function of time in Fig. 5g–i. The OOP magnetization is basically constant with time and no drift is observed. The in-plane magnetization manifests tiny effects in the RL with a frequency equal to the uniform FL droplet precession, which is 14 GHz. This means the droplet precesses at the theoretical lower bound, given by the Zeeman frequency ($f_Z = \gamma/2\pi\mu_0 H$). However, the frequency of a dissipative droplet is predicted to be a monotonically decreasing function of its diameter and to only reach the lower bound in the limit of infinite size[28]. Our droplet is not particularly large, and the low frequency must be attributed to interactions with the reference layer, which slow down the precession.

A droplet pair emerges as the field is kept constant and the current is increased. Figure 5c–j presents the characteristics of this state. Neither the FL droplet nor the RL droplet is stable with time. The evolution of the droplet volume is mirrored by the $M_z$-component (Fig 5j). Starting at $t = 1$ ns, the free layer droplet is fully developed, while the RL only demonstrates small wiggles. Similar wiggles are seen in the free layer. Figure 5c, d illustrates the presence of an FL droplet accompanied by a small RL perturbation. The reference layer excitation grows larger with time and after a certain interval the RL droplet starts to grow ($t \approx 4$ ns). Simultaneously, the FL droplet contracts. As the droplets reach about the same size (Fig. 5e, f), the FL droplet expands and the RL droplet diminishes. Once the RL droplet vanishes, new wiggles appear, and the process starts over.

The quasi-periodic DP state is thus characterized by an expansion/contraction process. It looks periodic at first glance, but the periodicity is far from perfect. Occasionally, the process is disrupted before the droplets reach their minimum/maximum size, see e.g., Fig. 5j at $t = 15$ ns. A stochastic element is clearly involved in the process. Moreover, the intervals between subsequential peaks in $M_z$ are not identical. The fast precessions of the $x$-components also exhibit both beating and a gradual phase shift. The stochastics must be driven by intrinsic non-linear dynamics governed by the underlying equations, and/or numerical noise, since $T = 0$ K rules out any thermal effects. Albeit unstable, the dynamics comprises different time scales that can

be estimated. The dominating in-plane precession frequency is similar to the single droplet state, $\approx 14$ GHz. The small wiggles in both components occur with a period of $\approx 0.45$ ns. The large peaks in $M_z$ emerge roughly every two nanoseconds, while the process is disrupted once in 10–15 ns.

A periodic DP state develops when the current is further increased (Fig. 6). Here, the free layer droplet is always accompanied by a reference layer counterpart. Each droplet gyrates with a frequency of $\approx 2.4$ GHz, around a point close to the NC center, and they expel each other. Their interdependence leads not only to a noticeable slowing down, $f = 12.7$ GHz, but also that both precess with nearly equal frequency. The sub-Zeeman frequency can be rationalized keeping the magnetic structure in mind. The periodic droplet pair exhibits very divergent in-plane textures (Fig. 6a, b), while a homogenous magnetic state is necessary to interpretate $f_Z$ as the lowest accessible frequency. The mutual precession frequency is on the other hand unexpected from single droplet theory. The layers have different anisotropy and saturation magnetization, but more importantly radically different degrees of spin reversal (mirrored by $M_z$ in Fig. 6i). The fact that the unequal properties do not affect the frequency, highlights the importance of the second droplet.

A chaotic state appears upon lowering both field and current ($\mu_0 H = 0.1$ T, $I = -6$ mA ($j = -1.2 \times 10^8$ A/cm$^2$)), as clearly seen in Fig. 6h–j. Both droplets are unstable and for short moments the FL-D practically vanishes, while the RL-D is missing for extended periods of time. There is no correlation between the $M_z$-components of the two droplets, in contrast to the quasi-periodic case. Besides, the in-plane precession is uniform only in the absence of an RL droplet.

The micromagnetic results compare well with the experiments, given the simple model. The stable FL droplet phase occur at low currents, while both the quasi-periodic and chaotic droplet pairs display substantial oscillations in $M_z$, which corresponds to an experimentally measurable signal. The oscillations appear on time scales on the order of several to tenths of nanoseconds, where the largest amplitudes are found for the longest cycles. These time scales match the experimental microwave maxima around 0.15 GHz, as well as the broad falloff towards higher frequencies.

The periodic droplet pair is not found in the experimental data, as this situation would result in a reduction/disappearance of the dynamic signal with current. Instead, we observe the biggest signals at the highest currents, which most likely is associated with large droplet drift. Then again, the simulations can not be expected to give a quantitative fit to all data. The model assumes zero temperature, and

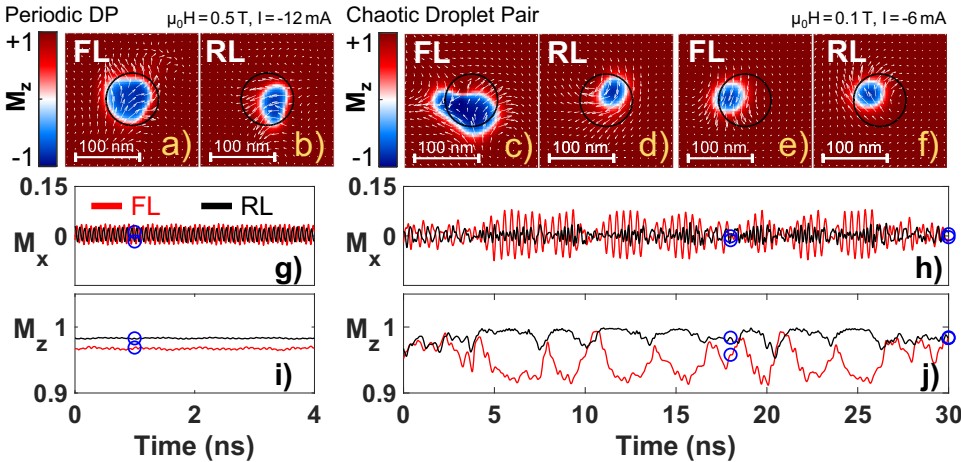

**Fig. 6 | Periodic and chaotic droplet pairs. a–f** Droplet snapshots together with **g–j** the time evolution of $M_x$ (in-plane) and $M_z$ (out-of-plane). The magnetization of the free layer (FL) and the reference layer (RL) are given by red and black lines, respectively. The blue circles in (**g–j**) show the corresponding time and magnetization of the snapshots.

at large currents this assumption certainly breaks down. Nevertheless, the simulations still reveal intriguing features of the coexisting droplets.

For completeness, we investigated the impact of finite temperatures by performing an additional simulation of the periodic droplet pair at $T = 300$ K. We have limited the high-temperature simulations to a single combination of field and current, since the calculations are extremely time-consuming. Already a run of 30 ns at $T = 0$ K takes about five days using a GPU cluster, and adding temperature effects makes the process three times slower. Figure S7 in the Supplementary Materials shows the time evolution of the droplet magnetization with and without temperature. The general behavior is preserved when temperature effects are included. The droplets still encircle each other, and the free layer droplet is consistently larger than the RL droplet. The precession frequency of 12.7 GHz is also unaffected. Nonetheless, the temperature does add random fluctuations to the dynamics. Hence, both droplets become more unstable, and the RL droplet occasionally shrinks and vanishes. These fluctuations give rise to low-frequency noise, consistent with experiments. Thus, the main consequence of finite temperatures is enhanced droplet drift, which should apply to all droplet pair scenarios described above.

### Concluding remarks

We conclude by pointing out the role of the reference layer, which has commonly been overlooked in studies of spin-torque stabilized magnetodynamical solitons. Our results show that the RL can not be neglected, particularly not for high currents. The experiments provide compelling evidence of coexisting droplets in the free and reference layer. This droplet soliton pair state is also reproduced in micromagnetic simulations. Furthermore, the simulations show that the droplet pair stability is tunable by field and current, which leads to various correlations between the RL- and FL droplets. The findings constitute a substantial step towards the comprehensive understanding of magnetic solitons needed to enable practical utilization of the phenomena. As a possible application, we here briefly mention that the strong incoherent broadband microwave noise, characteristic of the droplet pair state, might find use in so-called "radio lighting", i.e., the irradiation of surfaces, materials, and devices with incoherent broadband microwaves[56]. The current can switch on/off the droplet pair state on nanosecond time scales and further tune the upper cut-off of the microwave noise frequency band, which makes it a uniquely versatile source for incoherent microwave irradiation.

In addition, droplet pairs in STNOs offer a huge parameter space for experimental studies of interacting solitons, as well as their underlying non-linear equations. Beside the external handles – field, current, temperature – the two droplets reside in different media. It is thus straightforward to individually tailor the characteristics of each soliton. The mutual interaction is also easily tunable by interlayer exchange coupling (IEC), set by the Cu interlayer thickness. Consequently, the implications of the presented results go beyond plausible spintronic applications. The droplet soliton pair opens a new arena to explore the fundamentals of strongly non-linear phenomena.

## Methods
### Device fabrication
The multilayer stack, consisting of a seed layer, Ta (4)/Cu (14)/Ta (4)/Pd (2), an all-perpendicular pseudo-spin valve structure, [Co (0.35)/Pd (0.7)] × 5/Co (0.35)/Cu (5)/[Co (0.22) / Ni (0.68)] × 4/Co (0.22), and a cap layer, Cu (2)/Pd (2), was deposited on a thermally oxidized Si wafer using DC/RF magnetron sputtering (numbers in parentheses are thicknesses in nanometers). In the all-perpendicular pseudo-spin valve structure, the [Co/Pd] multilayer is regarded as the reference layer and the [Co/Ni] stack as the free layer, as presented in Fig. 1. Using conventional optical lithography and dry etching techniques, 8 µm × 16 µm mesas were patterned on the stacked wafer and insulated by a 30-nm-thick SiO₂ film using chemical vapor deposition (CVD). Then, electron beam lithography (EBL) and reactive ion etching (RIE) were used to fabricate nanocontacts through the SiO₂ on top of each mesa. The nanocontacts had a circular diameter varying from 50 to 150 nm. Finally, 500 nm of Cu followed by 100 nm Au was deposited on top and the contact electrode was produced by lift-off processing. The device used in the measurements displayed in Figs. 1–4 had a nanocontact with a 60-nm diameter.

### Magnetic and electrical characterization
The external fields were swept normal to the thin-film plane. Microwave and dc measurements of the fabricated STNOs were carried out using our custom-built 40-GHz probe station. It allows the manipulation of magnetic field strength, polarity, and direction. The device was connected through a ground–signal–ground (GSG) probe. The direct current, using a Keithley 6221 current source, flowed into the probe (so as the device) through a 40-GHz bias-Tee. The dc voltage was measured with a Keithley 2182 nanovoltmeter. Here we define the negative sign of the applied direct current as the electrons flow from the free to the reference layer. When the current generates enough STT,

auto-oscillation arises and emits a microwave signal. This microwave signal was decoupled from the dc voltage via the bias-Tee and then amplified using a low-noise amplifier prior to being recorded by a spectrum analyzer (R&S FSU 20 Hz–67 GHz). The bandpass width was 5 MHz.

## Micromagnetic simulations

Micromagnetic simulations were performed using the GPU-based open-source MuMax3 code[55]. Default settings were used for the solver, including the time step duration. The STNO was modeled by $512 \times 512 \times 3$ cells with a cell size of $3.90625 \times 3.90625 \times 3.90625$ nm$^3$. *Region1* was defined as the bottom $512 \times 512$ layer and corresponds to the [Co/Pd] reference layer. *Region2* constituted the top layer representing the [Co/Ni] free layer. The middle layer refers to the Cu spacer. The different thicknesses of the FL (RL) layer were accounted for by setting the variable "FreeLayerThickness.SetRegion" to 3.90625 nm for *Region2* (7.8125 nm for *Region1*).

The drive current flow was modeled by a simple cylinder with an 80-nm diameter (NC size) and the Oersted field was calculated and included. Zhang-Li torque was not taken into consideration in the simulations, since the current path was modeled without any xy-component. The potential impact of diffusive spin transport is discussed in Note 2 in the Supplementary Materials. The interlayer exchange coupling between the RL and FL was set to 0. Magnetic parameters of the FL (RL) were the uniaxial magnetic anisotropy $K_u = 340$ kJ/m$^3$ ($375$ kJ/m$^3$) as determined by out-of-plane FMR measurements, together with the literature value of the saturation magnetization $M_s = 716.2$ kA/m ($730$ kA/m)[57]. The same standard values were used for both layers: gyromagnetic ratio $\gamma/2\pi = 28$ GHz/T, exchange stiffness $A_{ex} = 10$ pJ/m, damping constant $\alpha = 0.03$, current polarization $P = 0.4$, and spin-torque asymmetry parameter $\Lambda = 1.3$. To mimic the back-hopping effect, we consider the real-time magnetization of one layer as the polarization layer of the other layer. In other words, the real-time states affect each other through the STT effect, and the polarization was updated for each time step.

Absorbing boundary conditions in the form of a smoothly increasing damping profile were applied to the simulated sample edges to avoid any interference artifacts from spin wave reflection. The applied field and initial magnetization angle were set to 89.7 degrees to mimic uncertainties in the experimental setup and to avoid any singularities associated with using an exact number of 90 degree. The magnetization components shown in Figs. 5g–j and 6g–j were calculated over an area of $128 \times 128$ cells, equivalent to $500 \times 500$ nm$^2$, with the nanocontact in the middle. The NC region thus constitutes about 2% of the sampled area. The momentary magnetization values were saved every 6 ps.

## Data availability

The data used to produce the plots within this paper are available at figshare.com [https://doi.org/10.6084/m9.figshare.25112297].

## Code availability

The MuMax3 code generated and analyzed during the current study is available at figshare.com [https://doi.org/10.6084/m9.figshare.25112297].

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

## Acknowledgements

We thank Artem Litvinenko for fruitful discussions on radio lighting. Support from the Swedish Research Council (VR; Dnr 2017-06711 and 2019-04229) (J. Å.) is gratefully acknowledged. S.J. acknowledges the financial support from the Natural Science Foundation of China (Grant 621044196) and Basic Research Programs of Taicang (Grant TC2021JC19), and Chongqing Natural Science Foundation (Grant 2022NSCQ- MSX4891). S.C. acknowledges support from the National Research Foundation of Korea (NRF) grant, funded by the Korean government (MSIT) (Grants 2022M3F3A2A03014536 and RS-2023-00244501). The work by O.H. was funded by the US Department of Energy Office of Science Basic Energy Sciences Division of Materials Science and Engineering. We gratefully acknowledge the computing resources provided on Blues and Bebop, high-performance computing clusters operated by the Laboratory Computing Resource Center at Argonne National Laboratory.

## Author contributions

S.J., S.C., Q.T.L., and H.M. characterized and optimized the material stack. S.J., S.C., Q.T.L., and A.H. and fabricated the devices. S.J. and S.C. performed all the microwave and electrical measurements. A.F. and O.H. carried out the micromagnetic simulations. S.C. and J. Å. initiated and supervised the project. All authors contributed to the data analysis and co-wrote the manuscript.

## Funding

## Competing interests

The authors declare no competing interests.
