## [Peer Review File · Nature Communications]

Reviewers' Comments:

Reviewer #1:

Remarks to the Author:

The manuscript reports current-induced effects in point-contact spin torque nano-oscillators (STNO) with perpendicular anisotropy. The new effect inferred from the analysis of the dc resistance and microwave spectra is the formation of a pair droplet state - a dynamical state that contains droplets in both the free and the reference layer of the nanocontact - at sufficiently high currents applied to the nanocontact. This interpretation is supported by the micromagnetic simulations. These findings help the understanding of complex current-induced static and dynamical magnetization states in magnetoelectronic nanostructures, which are relevant to their applications in information technology. Overall, the manuscript is well-written, the interpretation is supported by complementary measurements and simulations, and sufficient details are provided for other researchers to be able to reproduce these results. I have several suggestions for improving the clarity of the presentation, and a couple of questions about the results and interpretation/modelling.

- i) The third paragraph in the introduction discusses the importance of reference layer dynamics, without introducing the structure being discussed. Droplets have been observed into only in STNO, but also in SOT-driven structures that do not contain a reference layer, see e.g. PRB 96, 224419 (2017). Therefore, I think it is important to add a sentence briefly introducing the discussed types of structures and the reference layer.
- ii) The next paragraph starts with "We use nanocontact STNO..." and then describes their properties but does not say what purpose the STNO are used for. I think this paragraph would be a good place to introduce the scientific problem addressed by the study.
- iii) There is a misprint in the label for the cap layer in Fig.1(a). It says "Co/Pd" instead of "Cu/Pd". I was also slightly confused by this schematic, because it omits the electrode on the other side of the nanocontact that completes the circuit. Can this schematic be modified to minimize such confusion?
- iv) In my opinion, Figure 2 needs to be restructured to improve clarity. The difference between grey and white curves depicting the opposite current scan directions is barely discernible, making the corresponding arrows and the label confusing. The figure is also too crowded. This especially concerns panel (b) which contains essentially twice as much data as panel (a). The inset in panel (b) is too small and the labels are barely legible. I suggest that the panels/insets are re-arranged, and perhaps a separate panel is added to improve legibility.
- v) I feel that the use of "i.e." in the text is somewhat excessive. I suggest that it is replaced by synonymous expressions such as "as manifested by" or "implying that" etc.
- vi) In the resistance vs current curves, the parabolic background due to Joule heating was subtracted. It would be helpful if raw data without such subtraction are included as supplemental material.
- vii) The manuscript mentions Zhang-Li torque in passing as the explanation for asymmetry between results for opposite field directions. Many readers are likely unfamiliar with this mechanism. It would be helpful to briefly introduce it in one sentence, and provide a citation to the original paper(s).
- viii) The sentence "The behavior at negative currents is identical to $\mu_0 H = +0.2 \text{ T}$ " is confusing. I see that the noise spectra are similar, but there are two distinct resistance curves. The AP-state curve, which has been discussed until this point, is different from the positive-field curve. The curve for the P state matches the positive-field curve, but this is discussed only later in the paragraph.
- ix) I am confused by the discussion of rf emission near the bottom of page 2. For instance, it is stated that the power is diminished in all cases towards 5 GHz, but in the graphs I only see

gradual increase with increasing frequency, above 2GHz for positive current, and at all frequencies for negative current. Why did the authors pick 5 and not, say, 3GHz, and what is the significance of these features?

x) The lines outlining different regimes in Fig.3 are barely, if at all, discernible in some areas due to the similarity of colors. Is there a better choice of color scale or a different representation of the data?

xi) A reference is missing on top of page 6 (there is a "?" sign instead of a citation).

xii) I am confused by the units and the magnitudes shown in Fig.3b. My understanding is that the quantity measured by microwave spectroscopy is power spectral density (PSD), as shown e.g. in Figs. 2,4 . PSD has units of power per frequency band. If this is the quantity being integrated, aren't the units of the integral simply the power, not power (dB) multiplied by frequency? On the other hand, if one considers the positive-current DP spectra that are flat up to 2GHz, and then quickly decay, wouldn't the corresponding integral value of about 1000 dB*GHz as inferred from Fig.3b imply that the power emitted into a 1GHz band is ~50W?

xiii) The insets in Fig.4 are too small, the text is almost illegible.

xiv) I am confused by some aspects of the micromagnetic simulations. It is stated that "we considered the real-time magnetization of one layer to serve as the polarization layer of the other layer". The term "polarization layer" seems to be micromagnetic simulation jargon, and probably needs to be rephrased for the general audience. Do the simulations account for the diffusive nature of spin transport in the studied metallic structures, which can significantly affect the spin torque in highly inhomogeneous states? If not, can the authors estimate the significance of such effects? What is the significance of thermal effects also neglected in simulations, say, for the coherence of periodic droplet dynamics inferred from the simulations?

xv) Why does the experimentally observed DP appear at a much larger current than in the simulations?

xvi) It is emphasized in several places in the text that the coherent in-plane droplet dynamics does not produce coherent microwave signals due to the all-perpendicular geometry. Is it possible to slightly modify the experimental geometry, e.g. tilt the field, so that a finite coherent microwave signal due to precession can be observed? This would be very helpful as a confirmation for the presented interpretation and micromagnetic simulations, especially for the distinction between chaotic and quasi-periodic regimes of droplet dynamics revealed by simulations.

xvii) It would be helpful for the general audience if the conclusion could expand on the possible ways droplet pairs could appear in other geometries/structures and/or in applications of magnetic nanostructures.

Sincerely,
Sergei Urazhdin

Reviewer #2:

Remarks to the Author:

REPORT

on Ms. 23-29090 "Magnetic droplet soliton pairs" by S. Jiang et.al.

The manuscript reports the first experimental evidence for the formation of pairs of magnetic droplet solitons in two ferromagnetic multilayers with strong perpendicular magnetic anisotropy (PMA) separated by a nonmagnetic metal spacer. The solitons are induced by an electric current passing through such stack, which creates a spin-transfer torque acting on the magnetizations of the ferromagnetic Co/Ni and Co/Pd multilayers. Since the Co/Ni multilayer has smaller PMA than the Co/Pd one, the spin-transfer torque generates the soliton in the free Co/Ni layer at a lower

critical current density than in the reference Co/Pd layer (Fig. 2). By measuring the stack electrical resistance and the power spectral density, the authors determined magnetic states forming in the free and reference layers at various currents I and magnetic fields H and constructed the phase diagram of the studied material system in the I - H plane (Fig. 3). The diagram shows that the stability ranges of droplet pairs are limited by high currents and low magnetic fields. The experimental data also demonstrate the existence of an interaction between two solitons and a lower stability of the droplet pair in comparison with that of the single soliton appearing in the free layer. The formation of droplet pairs is confirmed by the predictions of micromagnetic simulations.

Although the results reported in the manuscript are interesting and noteworthy, the appearance of soliton pairs in the nanostructure comprising two ferromagnetic layers with PMA is not surprising. Indeed, the magnetizations of both layers experience the spin-transfer torque in the presence of a spin-polarized current so that two solitons may form when the current density is sufficiently high. Furthermore, the presentation of the results should be significantly improved to make the paper suitable for publication. The following changes are suggested:

- (i) The discussion of the electrically induced spin dynamics in the introduction is incomplete. Namely, spin waves also can be generated by the modulation of the voltage-controlled interfacial anisotropy in ferromagnet-dielectric nanostructures (Phys. Rev. Appl. 1, 044006, 2014; Appl. Phys. Lett. 111, 052404, 2017; Phys. Rev. B 104, 134422, 2021). This technique is distinguished by a low power consumption, because the anisotropy modulation is achieved by the application of a microwave voltage, but not an electric current with a high density, as required for the generation of spin dynamics by the spin-transfer and spin-orbit torques. Furthermore, it was predicted theoretically that even magnetic droplet solitons could be generated in the ferromagnet-dielectric nanostructures by short voltage pulses changing of the interfacial anisotropy without compensation of magnetic damping (Phys. Rev. Mater. 6, L101401, 2022).
- (ii) The authors characterize the appearance of droplets and droplet pairs only by values of the critical current (e.g., on pages 3 and 4). The critical current densities must be indicated and discussed in the paper as well.
- (iii) A reference is missing on page 6 (second line from the top).
- (iv) Micromagnetic simulations have been performed under the assumption of zero absolute temperature. It is important to check the stability of droplet pairs in the presence of thermal fluctuations by carrying out additional simulations accounting for a stochastic Gaussian noise.
- (v) The spin-torque nano-oscillators are promising for several practical applications. The authors should discuss how the formation of droplet pairs affects characteristics of nano-oscillators relevant to applications.

Mandatory revision is recommended.

GÖTEBORGS UNIVERSITET
INSTITUTIONEN FÖR FYSIK

We thank the reviewers for their questions and suggestions, which have helped us to improve the manuscript. We believe that we have addressed all their comments in an appropriate manner, and our response is found below, where we have listed all the remarks together with our answers.

Reviewer #1

The manuscript reports current-induced effects in point-contact spin torque nano-oscillators (STNO) with perpendicular anisotropy. The new effect inferred from the analysis of the dc resistance and microwave spectra is the formation of a pair droplet state - a dynamical state that contains droplets in both the free and the reference layer of the nanocontact – at sufficiently high currents applied to the nanocontact. This interpretation is supported by the micromagnetic simulations. These findings help the understanding of complex current-induced static and dynamical magnetization states in magnetoelectronic nanostructures, which are relevant to their applications in information technology. Overall, the manuscript is well-written, the interpretation is supported by complementary measurements and simulations, and sufficient details are provided for other researchers to be able to reproduce these results. I have several suggestions for improving the clarity of the presentation, and a couple of questions about the results and interpretation/modelling.

Question/Remark 1. *The third paragraph in the introduction discusses the importance of reference layer dynamics, without introducing the structure being discussed. Droplets have been observed into only in STNO, but also in SOT-driven structures that do not contain a reference layer, see e.g. PRB 96, 224419 (2017). Therefore, I think it is important to add a sentence briefly introducing the discussed types of structures and the reference layer.*

Answer 1. We thank the reviewer for this suggestion. To clarify the structure, we have added the following sentences:

“While droplets can be created in devices with a single ferromagnetic (FM) layer by utilizing the spin Hall effect [Divinskiy2017] and, in simulations, using voltage controlled magnetic anisotropy [Nikitchenko2022], they have mostly been examined using spin-torque nano-oscillators (STNOs), which contain two magnetic layers. In STNOs an electrical current run through a stack comprising a soft FM layer, a non-magnetic spacer, and a hard FM layer. The electrons are polarized by the hard *reference* layer, and the resulting spin current counteracts the damping in the soft *free* layer in which a droplet forms. Earlier studies have generally focused only on the easily excited magnetodynamics of the free layer, neglecting any dynamics in the reference layer.”

Q2. *The next paragraph starts with “We use nanocontact STNO...” and then describes their properties but does not say what purpose the STNO are used for. I think this paragraph would be a good place to introduce the scientific problem addressed by the study.*

GÖTEBORGS UNIVERSITET
INSTITUTIONEN FÖR FYSIK

A2. We thank the reviewer also for this suggestion. We have rephrased the paragraph, which now reads:

“We use nanocontact STNOs with strong perpendicular magnetic anisotropy (PMA) in both magnetic layers. The layer materials and stack order are shown in Fig. 1a. The device layout allows for the creation of a stable ordinary droplet in the free layer, illustrated in Fig. 1b. Furthermore, the all-perpendicular symmetry also opens up for droplet nucleation in the RL. In this study we apply large current densities to excite significant dynamics in the reference layer and investigate the limit of the RL acting as simply a static polarization layer. We observe clear transitions at high currents and show that both the RL and FL can sustain droplets, which coexist as depicted in Fig. 1c. The droplet pair constitutes a previously unexplored segment in the STNO current-field phase diagram, which we examine by experiments and simulations.”

Q3. *There is a misprint in the label for the cap layer in Fig.1(a). It says “Co/Pd” instead of “Cu/Pd”. I was also slightly confused by this schematic, because it omits the electrode on the other side of the nanocontact that completes the circuit. Can this schematic be modified to minimize such confusion?*

A3. We are glad that the reviewer noticed the misprint we missed. It is fixed now.

We understand the reviewer’s confusion, but are afraid that we might add new confusion, connected to length scales, if we add the top electrodes to the schematic in Fig. 1a. We want the nanocontact to be of similar size in Fig. 1a, b, and c. Instead, we have clarified the circuit by adding the following sentence to the caption:

“The applied current runs through the nanocontact down to the base electrode, where it flows laterally before it continues back up to two top electrodes (see Fig. S1 in the Supplementary Materials) located about 5 μm to the left and right of the NC.”

Q4. *In my opinion, Figure 2 needs to be restructured to improve clarity. The difference between grey and white curves depicting the opposite current scan directions is barely discernible, making the corresponding arrows and the label confusing. The figure is also too crowded. This especially concerns panel (b) which contains essentially twice as much data as panel (a). The inset in panel (b) is too small and the labels are barely legible. I suggest that the panels/insets are re-arranged, and perhaps a separate panel is added to improve legibility.*

A4. We have modified Figure 2 and hope the new version is clear. We use orange instead of gray for the back sweep and have removed the inset in panel (b). The purpose of the inset was to highlight small signals in the AP state, but we have chosen to add a Supplementary Figure in its place (Fig. S2).

GÖTEBORGS UNIVERSITET
INSTITUTIONEN FÖR FYSIK

Q5. *I feel that the use of “i.e.” in the text is somewhat excessive. I suggest that it is replaced by synonymous expressions such as “as manifested by” or “implying that” etc.*

A5. We have removed all “i.e.” but one.

Q6. *In the resistance vs current curves, the parabolic background due to Joule heating was subtracted. It would be helpful if raw data without such subtraction are included as supplemental material.*

A6. We agree and have added Supplementary Figures S4 and S5 showing the raw resistance data.

Q7. *The manuscript mentions Zhang-Li torque in passing as the explanation for asymmetry between results for opposite field directions. Many readers are likely unfamiliar with this mechanism. It would be helpful to briefly introduce it in one sentence, and provide a citation to the original paper(s).*

A7. We have added a short discussion on the mechanism:

“The difference in ΔR_D translates into a difference in size, and the smaller size of the AP-droplet is consistent with earlier results [Chung2018PRL]. The device geometry gives rise to a lateral current component [Banuazizi2017] that generates an in-plane Zhang-Li torque (ZLT) [Li2004, Zhang2004], which exerts inward (outward) pressure on the droplet perimeter for positive (negative) currents. Consequently, AP-droplets are smaller than their parallel state counterparts.”

Q8. *The sentence “The behavior at negative currents is identical to $\mu_0 H = +0.2 T$ ” is confusing. I see that the noise spectra are similar, but there are two distinct resistance curves. The AP-state curve, which has been discussed until this point, is different from the positive-field curve. The curve for the P state matches the positive-field curve, but this is discussed only later in the paragraph.*

A8. We understand that our original phrasing confused the reader. The opening of the paragraph now reads: “The anti-parallel state is stable for low negative currents, but at a threshold current the magnetic state switches from AP to P.”

Q9. *I am confused by the discussion of rf emission near the bottom of page 2. For instance, it is stated that the power is diminished in all cases towards 5 GHz, but in the graphs I only see*

GÖTEBORGS UNIVERSITET
INSTITUTIONEN FÖR FYSIK

gradual increase with increasing frequency, above 2GHz for positive current, and at all frequencies for negative current. Why did the authors pick 5 and not, say, 3GHz, and what is the significance of these features?

A9. Our old discussion was imprecise and is now hopefully improved. We have also added a Supplementary Figure (Fig. S3) presenting example slices of the colormap plot, which we believe will help the reader to compare the PSD of the two current polarities.

Per se, there is nothing special about 5 GHz, but towards this frequency the power of both conditions (P and AP) is similar and declining for all currents, making it a suitable reference. The similar signals imply that the dynamics of the P- and AP-droplet pairs are comparable on short timescales (large frequencies).

The new version of the discussion reads:

“The shape of the signal is also different. Examples are found in Fig. S3 in the Supplementary Materials. At negative currents, the power peaks at low f and gradually levels off for frequencies above the peak. The maximum moves towards higher f with increasing current. At positive currents, the signal shows a peak around 0.1 GHz for all currents above ~ 10 mA. At higher currents, $I < 15$ mA, the power is comparable to the initial peak for a range of frequencies, while above roughly 2 GHz the signal progressively weakens. Consequently, the drift motion of coexisting droplets depends on the initial orientation of the magnetic layers. Nevertheless, the signal of both conditions is similar when the frequency approaches 5 GHz, which imply comparable dynamics on the shortest time scales (largest f).

Q10. *The lines outlining different regimes in Fig.3 are barely, if at all, discernible in some areas due to the similarity of colors. Is there a better choice of color scale or a different representation of the data?*

A10. We have changed the color of the lines and increased their thickness. We also changed the colormap of Fig.3a. We hope the different regimes are discernible in this new version.

Q11. *A reference is missing on top of page 6 (there is a “?” sign instead of a citation).*

A11. We have fixed the misprint. The reference is Statuto et al., Multiple magnetic droplet soliton modes, Phys. Rev. B **99**, 174436 (2019).

Q12. *I am confused by the units and the magnitudes shown in Fig.3b. My understanding is that the quantity measured by microwave spectroscopy is power spectral density (PSD), as shown e.g. in Figs. 2,4 . PSD has units of power per frequency band. If this is the quantity being*

GÖTEBORGS UNIVERSITET
INSTITUTIONEN FÖR FYSIK

*integrated, aren't the units of the integral simply the power, not power (dB) multiplied by frequency? On the other hand, if one considers the positive-current DP spectra that are flat up to 2GHz, and then quickly decay, wouldn't the corresponding integral value of about 1000 dB*GHz as inferred from Fig.3b imply that the power emitted into a 1GHz band is ~50W?*

A12. We are very grateful that the reviewer noted this inconsistency. The presented units were indeed incorrect. The correct unit of the measured PSD is dB over the noise floor. To get proper units for the integrated PSD one needs to convert the signal to W/Hz. However, a correct integration procedure gives a result where the small signals of the single droplet mode transitions vanish in the DP/drift background, making it impossible to visualize all interesting features of the individual spectra in a single plot.

Since the purpose of the figure is to highlight characteristics of the different states and produce a phase diagram, we have chosen to keep the somewhat unphysical approach of summing up decibels. We have commented the implications of this approach in the first paragraph of the section 'Current-field phase diagram':

"The integration is made by summing up the measured power in dB (over the noise floor). Consequently, the relation to true power is lost, and the unit is undefined. Nevertheless, this approach makes it possible to display small and large signals within the same plot, whereas minor peaks are indistinguishable if the linear scale power [W/Hz] is integrated."

Q13. *The insets in Fig.4 are too small, the text is almost illegible.*

A13. It is hard to fit larger insets in Fig. 4, but we have added magnified versions of the insets in the Supplementary Materials (Fig. S6).

Q14. *I am confused by some aspects of the micromagnetic simulations. It is stated that "we considered the real-time magnetization of one layer to serve as the polarization layer of the other layer". The term "polarization layer" seems to be micromagnetic simulation jargon, and probably needs to be rephrased for the general audience. Do the simulations account for the diffusive nature of spin transport in the studied metallic structures, which can significantly affect the spin torque in highly inhomogeneous states? If not, can the authors estimate the significance of such effects? What is the significance of thermal effects also neglected in simulations, say, for the coherence of periodic droplet dynamics inferred from the simulations?*

A14. The question contains three sub-questions on i) the term "polarization layer", ii) the effect of diffusive spin currents, and iii) the significance of thermal effects. We respond to each sub-question individually.

GÖTEBORGS UNIVERSITET
INSTITUTIONEN FÖR FYSIK

i) The term “polarization layer” is a synonym to the more frequently used terms “fixed layer” or “reference layer”. We believe that most readers are familiar with the meaning of the terms in a STNO context, but we acknowledge that our original phrasing was difficult to comprehend. We have adjusted the remarked sentence and hope that the new version is understandable:

“A drawback of the used MuMax3 package [Vansteenkiste2014] is that the polarization (fixed) layer magnetization is constant by default, which means that back-hopping effects are ignored. To include these effects, we considered the real-time magnetization of one layer to serve as the polarization layer of the other layer.”

ii) To our knowledge, the simulation only calculates the spin polarization created within the current flow, which is modeled by a simple cylindrical shape of NC radius. Additional diffusive currents originating from spin accumulation outside of this boundary is not accounted for. We expect that the major mechanisms are captured within the present model, since the impact of the RL/FL magnetic state on the Slonczewski torque is covered and the greater part of the spin current stem from the NC area. Any additional effects should be of less importance. However, as an example, we do know that the excluded Zhang-Li torque affects the droplets size. Hence, the simulations are not to be considered as a one-to-one representation of the experimental system, but a model that accounts for the most important physical mechanisms.

iii) We have performed an additional simulation including thermal effects, setting $T = 300$ K. The results show that the main effect of the thermal field is to add random fluctuations to the dynamics, which cause more unstable droplet states. However, the main features are preserved, and we conclude that the $T = 0$ K simulations indeed unveil relevant scenarios.

We have added a Supplementary Figure, Fig. S7, and included the following discussion in the manuscript:

“For completeness, we investigated the impact of finite temperatures by performing an additional simulation of the periodic droplet pair at $T = 300$ K. We have limited the high temperature simulations to a single combination of field and current, since the calculations are extremely time consuming. Already a run of 30 ns at $T = 0$ K takes about five days using a GPU cluster, and adding temperature effects makes the process three times slower. Figure S7 in the Supplementary Materials shows the time evolution of the droplet magnetization with and without temperature. The general behavior is preserved when temperature effects are included. The droplets still encircle each other, and the free layer droplet is consistently larger than the RL droplet. The precession frequency of 12.7 GHz is also unaffected. Nonetheless, the temperature does add random fluctuations to the dynamics. Hence, both droplets become more unstable, and the RL droplet occasionally shrinks and vanishes. These fluctuations give rise to low frequency noise, consistent with experiments. Thus, the main consequence of finite temperatures is enhanced droplet drift, which should apply to all droplet pair scenarios described above.”

GÖTEBORGS UNIVERSITET
INSTITUTIONEN FÖR FYSIK

Q15. *Why does the experimentally observed DP appear at a much larger current than in the simulations?*

A15. We don't expect the simulations to quantitatively reproduce the experimental data. The nucleation current has a complex dependence on a variety of parameters, such as the magnetic properties, the damping, the current polarization efficiency, the current distribution, temperature, etc. It is usually possible to explore this parameter space to get a better fit to experiments, but in our case each simulation takes about a week of computing time, making exploration impossible in practice (or at least extremely time consuming). Therefore, we have reused values from former successful single droplet simulations, added reasonable estimates for the reference layer and focused on the general behavior.

Q16. *It is emphasized in several places in the text that the coherent in-plane droplet dynamics does not produce coherent microwave signals due to the all-perpendicular geometry. Is it possible to slightly modify the experimental geometry, e.g. tilt the field, so that a finite coherent microwave signal due to precession can be observed? This would be very helpful as a confirmation for the presented interpretation and micromagnetic simulations, especially for the distinction between chaotic and quasi-periodic regimes of droplet dynamics revealed by simulations.*

A16. It is indeed possible to tilt the field to get a signal from the in-plane dynamics. We have used this in earlier publications to confirm the presence of single droplets in all-perpendicular devices [Chung et al., PRL 120, 217204 (2018), Ahlberg et al., Nature Com., 13, 2462 (2022)]. However, it is less straightforward to use this kind of data to explore the nature of the droplet pairs. A careful analysis is needed to distinguish the different states. Probably the best method is to record time traces of the resistance, instead of an average signal on the MHz-GHz scale, but we have currently no such data.

Q17. *It would be helpful for the general audience if the conclusion could expand on the possible ways droplet pairs could appear in other geometries/structures and/or in applications of magnetic nanostructures.*

A17. While the interest in magnetodynamic droplet solitons and droplet pairs is primarily curiosity-driven, it is interesting to think about how they could be useful for applications. The most striking feature of the droplet pair state is the appearance of strong, wideband microwave noise. While this might seem like a nuisance, there exist applications of so-called "radio lighting" where incoherent broadband microwave sources are used to irradiate a surface, material, or device. Such sources are surprisingly uncommon. As the droplet pair state is both current and field tunable and can be switched on/off at nanosecond time scales, an STNO capable of generating droplet pairs could be a versatile source for incoherent broadband

GÖTEBORGS UNIVERSITET INSTITUTIONEN FÖR FYSIK

microwaves. From Fig.4(a)-(d), we can conclude that at low field (0.35 T), the current can be used to switch on/off the source and also tune its upper cut-off frequency up to 2 GHz. At higher field (0.5 T), there also exists a lower cut-off frequency for the broadband noise. Through a combination of current and field, one can hence tune both the upper and the lower cut-off frequency. We have added two sentences to the conclusion where we briefly mention this application, including a reference to “radio lighting”.

“As a possible application, we here briefly mention that the strong incoherent broadband microwave noise, characteristic of the droplet pair state, might find use in so-called "radio lighting", i.e. the irradiation of surfaces, materials, and devices with incoherent broadband microwaves. [Dmitriev2016] The current can switch on/off the droplet pair state on nanosecond time scales and further tune the upper cut-off of the microwave noise frequency band, which makes it a uniquely versatile source for incoherent microwave irradiation.”

Reviewer #2

REPORT

on Ms. 23-29090 “Magnetic droplet soliton pairs” by S. Jiang et.al.

The manuscript reports the first experimental evidence for the formation of pairs of magnetic droplet solitons in two ferromagnetic multilayers with strong perpendicular magnetic anisotropy (PMA) separated by a nonmagnetic metal spacer. The solitons are induced by an electric current passing through such stack, which creates a spin-transfer torque acting on the magnetizations of the ferromagnetic Co/Ni and Co/Pd multilayers. Since the Co/Ni multilayer has smaller PMA than the Co/Pd one, the spin-transfer torque generates the soliton in the free Co/Ni layer at a lower critical current density than in the reference Co/Pd layer (Fig. 2). By measuring the stack electrical resistance and the power spectral density, the authors determined magnetic states forming in the free and reference layers at various currents I and magnetic fields H and constructed the phase diagram of the studied material system in the I - H plane (Fig. 3). The diagram shows that the stability ranges of droplet pairs are limited by high currents and low magnetic fields. The experimental data also demonstrate the existence of an interaction between two solitons and a lower stability of the droplet pair in comparison with that of the single soliton appearing in the free layer. The formation of droplet pairs is confirmed by the predictions of micromagnetic simulations.

Although the results reported in the manuscript are interesting and noteworthy, the appearance of soliton pairs in the nanostructure comprising two ferromagnetic layers with PMA is not surprising. Indeed, the magnetizations of both layers experience the spin-transfer torque in the presence of a spin-polarized current so that two solitons may form when the current density is sufficiently high. Furthermore, the presentation of the results should be significantly improved to make the paper suitable for publication. The following changes are suggested:

GÖTEBORGS UNIVERSITET
INSTITUTIONEN FÖR FYSIK

Question/Remark 1. *The discussion of the electrically induced spin dynamics in the introduction is incomplete. Namely, spin waves also can be generated by the modulation of the voltage-controlled interfacial anisotropy in ferromagnet-dielectric nanostructures (Phys. Rev. Appl. 1, 044006, 2014; Appl. Phys. Lett. 111, 052404, 2017; Phys. Rev. B 104, 134422, 2021). This technique is distinguished by a low power consumption, because the anisotropy modulation is achieved by the application of a microwave voltage, but not an electric current with a high density, as required for the generation of spin dynamics by the spin-transfer and spin-orbit torques. Furthermore, it was predicted theoretically that even magnetic droplet solitons could be generated in the ferromagnet-dielectric nanostructures by short voltage pulses changing of the interfacial anisotropy without compensation of magnetic damping (Phys. Rev. Mater. 6, L101401, 2022).*

A1. We thank the reviewer for reminding us of these references and the complementary technique to generate droplets. To include this perspective, we have added the first three references to the second sentence of the introduction, together with the added text:

“...and/or voltage controlled magnetic anisotropy (VCMA) in spintronic nanodevices”

To include the concept of VCMA created droplets we have added the following sentence further down in the introduction:

“While droplets can be created in devices with a single ferromagnetic (FM) layer by utilizing the spin Hall effect [Divinskiy2017] and, in simulations, using voltage controlled magnetic anisotropy [Nikitchenko2022], they have mostly been examined using spin-torque nano-oscillators, which contain two magnetic layers.”

Q2. *The authors characterize the appearance of droplets and droplet pairs only by values of the critical current (e.g., on pages 3 and 4). The critical current densities must be indicated and discussed in the paper as well.*

A2. We have added current density values to the manuscript.

Q3. *A reference is missing on page 6 (second line from the top).*

A3. We have fixed the misprint. The reference is Statuto et al., Multiple magnetic droplet soliton modes, Phys. Rev. B 99, 174436 (2019).

Q4. *Micromagnetic simulations have been performed under the assumption of zero absolute temperature. It is important to check the stability of droplet pairs in the presence of thermal fluctuations by carrying out additional simulations accounting for a stochastic Gaussian noise.*

GÖTEBORGS UNIVERSITET
INSTITUTIONEN FÖR FYSIK

A4. This remark is very similar to Q14(iii) by Reviewer #1, and we kindly refer to that answer for the full response. As we state above, we have made an additional simulation. The results show that finite temperatures add random fluctuations to the magnetodynamics, as expected, and thus the droplets become more unstable. But most importantly the general behavior is preserved and the 0 K simulations do give an insight to the different states of a droplet pair.

Q5. *The spin-torque nano-oscillators are promising for several practical applications. The authors should discuss how the formation of droplet pairs affects characteristics of nano-oscillators relevant to applications.*

A5. As mentioned in our answer to Q17 by the first reviewer (please see that answer for details), the strong incoherent broadband microwave noise can be used for so-called “radio lighting”. We have added two sentences to the conclusion where we briefly mention this application, including a reference.

Reviewers' Comments:

Reviewer #1:

Remarks to the Author:

The authors adequately addressed most of my comments/suggestions, with a couple of exceptions. I believe these minor issues can be easily addressed by the authors without the need for a further review.

1. In response to my comment labeled Q9 in the rebuttal letter, the description of the observed microwave spectra vs bias was revised. Unfortunately, I still find the revised version confusing. For example, I find the statement "the power peaks at low f " too imprecise. The term "levels off" in reference to the emitted power at higher frequencies seems to imply that it approaches a finite constant value, but it seems to decay. I find the revised discussion of emission at positive current also confusing. The authors end the paragraph by stating that "Consequently, the drift motion of coexisting droplets depends on the initial orientation of the magnetic layers". I was not able to follow how this statement was inferred from the previous ones. In the next sentence, "imply" should be probably "implies", but aside from this, there is not much signal at 5GHz, so I am lost as to how one can infer similarity of short timescale dynamics from the absence of signals.

2. I have some technical concerns about the changes made in response the comment numbered by the authors Q12. I understand that spectrum analyzers can measure power in dB. For peaks with the width smaller than bandpass filter, this value represents the total power of the signal. However, for broad spectral distributions, the measured value in dB depends on the width of the bandpass filter. In other words, the unit of measurement is undefined as correctly noted by the authors. However, in this case even the amplitude of the signal becomes dependent on the bandpass width. Is it possible to integrate the actual power spectral density (not its logarithm) and then use logarithmic scale to highlight the small signals? If not, the bandpass width needs to be specified to make the results reproducible by others.

3. My question Q3 part ii), as labeled by the authors, was about the effect of inhomogeneity of the magnetization state in the polarizer/free layer on spin polarization. I apologize that this question may have been phrased in a confusing manner. The response states that the effects of magnetization state outside the nanocontact region are not included, and mentions the in-plane Zhang-Li torque. However, my question was more about the effects of magnetic inhomogeneity of , say, polarizer, on spin polarization experienced by the free layer due to diffusive spin transport (i.e. spin distribution described by 3d drift-diffusion equation). My understanding is that the model simply maps the spin polarization from the magnetization locally along the axis of NC. It would be nice if an estimate of how relevant such lateral diffusion effects may be to the considered highly inhomogeneous states.

4. I would appreciate if the response to Q15, an explanation of the quantitative differences between experiments and simulations, is included in some form, perhaps in the supplemental text.

I have reviewed the comments of the second reviewer and found the responses provided by the authors adequate.

Sincerely,
Sergei Urazhdin

Reviewer #2:

Remarks to the Author:

The authors properly addressed the critical remarks formulated by the reviewers and significantly improved the presentation of their results. I especially appreciate additional micromagnetic simulations of the magnetization dynamics at room temperature and the idea of using pairs of magnetic droplet solitons for applications in the so-called "radio lighting". The paper may be

published as is.

GÖTEBORGS UNIVERSITET INSTITUTIONEN FÖR FYSIK

We thank the reviewers for input, which have helped us to improve the manuscript. We believe that we have addressed all comments in an appropriate manner, and our response is found below, where we have listed all the remarks together with our answers.

Reviewer #1 (Remarks to the Author):

Remark 0. *The authors adequately addressed most of my comments/suggestions, with a couple of exceptions. I believe these minor issues can be easily addressed by the authors without the need for a further review.*

Answer 0. We thank the reviewer for his comments, and that he trusts us with answering the remarks without further review.

R1. *In response to my comment labeled Q9 in the rebuttal letter, the description of the observed microwave spectra vs bias was revised. Unfortunately, I still find the revised version confusing. For example, I find the statement “the power peaks at low f ” too imprecise. The term “levels off” in reference to the emitted power at higher frequencies seems to imply that it approaches a finite constant value, but it seems to decay. I find the revised discussion of emission at positive current also confusing. The authors end the paragraph by stating that “Consequently, the drift motion of coexisting droplets depends on the initial orientation of the magnetic layers”. I was not able to follow how this statement was inferred from the previous ones. In the next sentence, “imply” should be probably “implies”, but aside from this, there is not much signal at 5GHz, so I am lost as to how one can infer similarity of short timescale dynamics from the absence of signals.*

A1. We have removed the incorrect use of “level off” and specified the frequency range where the power peaks. The sentence now reads: “At negative currents, the power peaks at low f (0–1 GHz) and decreases with frequency above the peak.”

We have also clarified our conclusions in the end of the paragraph, i.e. “Consequently, the drift motion of coexisting droplets depends on the initial orientation of the magnetic layers.”, and “Nevertheless, the signal of both conditions is similar when the frequency approaches 5 GHz, which imply comparable dynamics on the shortest time scales (largest f).” We have added the following text “In short, the signal shape of the droplet pair is different for the P and AP states, and the signal is determined by the DP motion.”, and rephrased the second statement to: “the signal of both conditions is similar for frequencies above ~ 3 GHz”.

We have corrected “imply” to “implies”.

R2. *I have some technical concerns about the changes made in response the comment numbered by the authors Q12. I understand that spectrum analyzers can measure power in*

GÖTEBORGS UNIVERSITET
INSTITUTIONEN FÖR FYSIK

dB. For peaks with the width smaller than bandpass filter, this value represents the total power of the signal. However, for broad spectral distributions, the measured value in dB depends on the width of the bandpass filter. In other words, the unit of measurement is undefined as correctly noted by the authors. However, in this case even the amplitude of the signal becomes dependent on the bandpass width. Is it possible to integrate the actual power spectral density (not its logarithm) and then use logarithmic scale to highlight the small signals? If not, the bandpass width needs to be specified to make the results reproducible by others.

A2. We have tried to integrate the actual power spectral density and then plot the result on a logarithmic scale, but this approach makes it impossible to visualize all features of the phase diagram in a single figure. The small signals disappear. Instead, we have specified the bandpass width in the Methods section; “*The bandpass width was 5 MHz.*”

R3. *My question Q3 part ii), as labeled by the authors, was about the effect of inhomogeneity of the magnetization state in the polarizer/free layer on spin polarization. I apologize that this question may have been phrased in a confusing manner. The response states that the effects of magnetization state outside the nanocontact region are not included, and mentions the in-plane Zhang-Li torque. However, my question was more about the effects of magnetic inhomogeneity of, say, polarizer, on spin polarization experienced by the free layer due to diffusive spin transport (i.e. spin distribution described by 3d drift-diffusion equation). My understanding is that the model simply maps the spin polarization from the magnetization locally along the axis of NC. It would be nice if an estimate of how relevant such lateral diffusion effects may be to the considered highly inhomogeneous states.*

A3. We have the same understanding as the reviewer; that the model simply maps the spin polarization from the magnetization locally along the axis of NC. We do believe that this captures the most relevant spin polarization/torques. Unfortunately, we don't have any estimates of the effect of potential spin diffusion, but expect the effect to be small.

R4. *I would appreciate if the response to Q15, an explanation of the quantitative differences between experiments and simulations, is included in some form, perhaps in the supplemental text.*

A4. We have added a supplementary note about the correspondence between the data and the simulations:

Supplementary note 1. On the correspondence between data and simulations.

GÖTEBORGS UNIVERSITET
INSTITUTIONEN FÖR FYSIK

The droplet pair nucleation current is larger in the experiments than in the simulations. We don't expect the simulations to quantitatively reproduce the experimental data. The nucleation current has a complex dependence on a variety of parameters, such as the magnetic properties, the damping, the current polarization efficiency, the current distribution, temperature, etc. It is usually possible to explore this parameter space to get a better fit to experiments, but in our case each simulation takes about a week of computing time, making exploration impossible in practice (or at least extremely time consuming). Therefore, we have reused values from former successful single droplet simulations, added reasonable estimates for the reference layer and focused on the general behavior.

In addition, the current distribution is simplified to a cylindrical flow, neglecting any in-plane current components and associated torques. Diffusive spin currents originating from the highly inhomogeneous magnetization of the polarizing layer are also omitted in the model, which only accounts for the spin polarization produced locally underneath the NC. Nonetheless, we expect that our uncomplicated model still captures the essentials of droplet pair dynamics, and that a more elaborate model would not show significantly different results.

Reviewer #2 (Remarks to the Author):

R0. *The authors properly addressed the critical remarks formulated by the reviewers and significantly improved the presentation of their results. I especially appreciate additional micromagnetic simulations of the magnetization dynamics at room temperature and the idea of using pairs of magnetic droplet solitons for applications in the so-called "radio lighting". The paper may be published as is.*

A0. We once again thank the reviewer for his/her efforts and are happy with the advice to publish as is.